# Game Analysis of the Open-Source Innovation Benefits of Two Enterprises from the Perspective of Product Homogenization and the Enterprise Strength Gap

**Aiping Tao [1], Qi Qi [1],\*, Yi Li [2],\*, Dan Da [3], Valentina Boamah [4] and Decai Tang [4]**

[1]  School of Economics, Hefei University of Technology, Hefei 230601, China; aipingtao@yeah.net
[2]  Jiangsu Office, China Banking and Insurance Regulatory Commission, Nanjing 210004, China
[3]  School of Business, Jiangsu Open University, Nanjing 210000, China; dadan@jsou.edu.cn
[4]  School of Management Science and Engineering, Nanjing University of Information Science & Technology, Nanjing 210044, China; 20215242005@nuist.edu.cn (V.B.); tangdecai@nuist.edu.cn (D.T.)
\*  Correspondence: qiqijsnj@163.com (Q.Q.); tolunto@126.com (Y.L.)

**Abstract:** Revenue is one of the hottest topics in the field of open-source innovation. Can open-source innovation really bring more revenue to firms? What affects the revenue from open-source innovation? Based on the perspective of product homogenization and the enterprise-strength gap, these questions are answered in this study using theoretical analyses and the construction of a game model to explore the influence of product homogeneity and the strength gap between firms regarding the revenue from open-source innovation. The results show that enterprise homogeneity and the revenue from open-source innovation are not linearly related. High homogeneity does harm the revenue from open-source innovation, while the revenue is relatively high when the homogeneity is moderate. Additionally, it was also identified that the strength gap between firms has a negative influence on the revenue from open-source innovation. The wider the strength gap is, the greater the revenue loss of the weaker firms and, thus, the lower the total revenue of the two firms will be. This paper provides a reference for research on enterprise revenue from open-source innovation and the selection of participants in open-source activities.

**Keywords:** open-source innovation; homogeneity; competitive intensity; knowledge spillover





## 1. Introduction

In the era of globalization and the Internet economy, technology and information are interacting more and more frequently. The extent of openness and the sharing of knowledge, technology, and other scientific and technological innovation results are increasing. The traditional "closed source innovation" inside firms is difficult to adapt to business development and competition [1], and gradually shows a certain inadaptability. Open-source innovation is becoming more and more popular because of its openness and the sharing of results. As early as 1999, the Linux open-source operating system was being developed; it has since become the biggest competitor of the Microsoft operating system, and has been encroaching on the market share of the Microsoft Windows system [2]. Apache, the open-source network software, has maintained a kingpin position in the market since its release in 2002, occupying more than 60% of the market of network server software in 2007. Nowadays, open-source technology is widely used in more than 80% of global software sales, and open-source technology supports more than 90% of scientific and technological innovation products [3]. Therefore, the importance of studying the benefits of open-source innovation is becoming increasingly evident. The most significant difference between open-source innovation and closed source innovation is the high degree of openness of the innovation results. The open innovation results are not closed and monopolized but are openly shared, allowing other firms to learn from them. Within the scope of the open-source licensing agreement, participants are free to participate in the innovation process

and produce open-source results and are also allowed to revise and disseminate them [4]. Every open-source innovator is, consequently, also an innovator of his predecessors' work. In Newton's words, they are "standing on the shoulders of giants" [5].

On the one hand, large enterprises choose open-source innovation to integrate resources and reduce costs so as to improve innovation capability [6–8]. On the other hand, they hope to obtain the first-mover advantages of technological innovation and establish technical standards [9]. Through open-source innovation, small enterprises can make better use of the external resources owned by their partners, develop diversified products and enhance competitiveness [10].

Although open-source firms lose the monopoly revenue garnered by closed-source innovation, they gain additional revenue by learning from each other's innovations. Whether open-source innovation can bring revenue to firms, and what factors influence the size of that revenue are important issues in the new field of open-source innovation. However, due to the many factors affecting the benefits of open-source innovation, there are few studies on its quantitative analysis through a mathematical model or empirical analysis. The uncertainty of revenue seriously hinders the promotion and secondary innovation of the open-source innovation model. A demonstration is urgently needed to show that open-source innovation can achieve revenue growth.

This paper analyzes the revenue channels of the open-source innovation model and builds a revenue function with the help of a game model. In this paper, the influencing factors of various forms of open-source innovation revenue are expressed using two variables: the enterprise product homogenization level and the enterprise-strength gap. In this way, the influence of enterprise homogenization level and the strength gap on open-source innovation revenue can be explored more clearly. The conclusions of this study can provide inspiration for the design of an open-source innovation incentive mechanism and the relevant policymaking of open-source community participants.

## 2. Literature Review

### 2.1. The Concept and Characteristics of Open-Source Innovation

Eric Raymond was the first to propose the term "open-source". In a narrow sense, open-source refers specifically to the source code of open software, to ensure the openness, accessibility, and easy modification of that source code. With the development of Internet technology and the deepening of the Internet economy, the term "open-source" has become more and more widely applied. The connotation of "open-source" has expanded and has gradually evolved into a kind of thought and culture that is mainly characterized by openness, freedom, and sharing [11].

Open-source innovation is an innovation method derived from open-source. Scholars of open-source innovation have different opinions. Lerner and Tirole define open-source innovation as a joint innovation of regional spatial dispersion with knowledge-intensive and interactive technicians and users through the Internet [9]. Hippel and Krogh define open-source innovation as a user-driven individual–collective joint innovation model [12]. Steve Hamm summarizes open-source innovation as a collaborative development process that includes multiple participants (contributors) [13]. It can be seen that multi-subject collaborative innovation and achievement-sharing are the most important core factor of open-source innovation.

Open-source innovation is mainly characterized by four aspects:

(1) Innovation subject diversification. The information technology attribute of the Internet breaks through the regional restrictions of traditional innovation. It enables the participants of open-source innovation to cooperate and innovate across regions and industries. Simon Grand and Georg von Kroghetc have pointed out that open-source innovation can combine the forces of individual developers and developers from different sectors, firms, and borders and "collaborinnovate" [6].

(2) Diversified incentive mechanisms. It is important to design a diversified incentive mechanism based on the diverse participation motivation of open-source innovation partic-

ipants. Participatory incentives include the fun of innovation, altruism, the signal effect of career promotion, etc., while outcome incentives include meeting the customized needs of innovation results and business benefits [11].

(3) Openness of the innovation process. Open-source innovation is typified by open innovation with high levels of openness in terms of both innovation resources and processes. The openness of the process is reflected in the fact that the knowledge and technology used in the process of innovation and development are no longer closed but are instead partially or even completely open. Firms can source inspiration and ideas from the knowledge and technology disclosed by other participants. They can also cooperate to integrate the technical strength of various firms to reduce the cost of innovation. That is to say, they realize the innovation process of wide-ranging coordination with low thresholds.

(4) Sharing of innovation achievements. Sharing is not only the concept of open-source innovation but also an important feature that distinguishes open-source innovation from other innovation models. Sharing avoids the wasting of resources caused by repeated parallel development and maximizes the economic and social value of innovative results. The sharing of open-source innovation results is reflected in two parts: source achievement sharing and secondary innovation achievement sharing. Open-source firms will publicly disclose the source results. All participants can refer to it and share their knowledge and technology. Then, some participants will optimize and improve the source results after learning from them, to form a secondary innovation. Via this optimizing and upgrading by participants, the sharing of secondary innovation results will be realized again.

In the process of open-source innovation, open-source enterprises develop source innovation results and obtain innovation benefits. Due to market competitiveness, other enterprises lose part of their profits. However, other companies have learned from the open-source innovation results (known as a "learning enterprise"). These enterprises gain new knowledge and new technology by which to increase profits. This process is also known as a high degree of knowledge and technology diffusion. In this diffusion process of knowledge and technology, the technology gap between learning enterprises and open-source enterprises decreases. The relative advantage of open-source enterprises inevitably costs part of the revenue. Next, the learning enterprise will choose whether to make a secondary innovation based on the source results and disclose their findings. Finally, if the learning enterprise chooses this secondary innovation, the open-source enterprise will learn the results and gain in terms of profits.

## 2.2. Open-Source Innovation Advantages and Its Source of Innovation Power

Open-source innovation extends the value of innovation results to a greater extent than closed-source innovation. The idea that open-source innovation can bring greater benefits to companies is increasingly widely accepted. Yu and Ding believe that an open-source license enables developers to use the relevant knowledge and technology without needing to master the relevant property rights and patent rights. This can significantly reduce knowledge and technology's circulation and transaction costs, thus contributing to revenue improvement overall [14]. Henkel believes that the openness of open-source innovation will lead to changes in the external environment and consumer demand. This change will increase the learning revenue that enterprise managers accrue from open-source innovation [15].

Currently, however, companies are being driven by profit alone. Hitchhiking problems are also common. How do we ensure that innovation power is the core issue of an open-source innovation model? Scholars have some optimism and some concerns about this issue. West and Gallagher answer one question of open-source innovation: "Why do companies have to invest in research and development if the results can be delivered to competitors through open-source innovation cases?" It is also pointed out that open innovation can help enterprises to effectively integrate internal and external resources, fully tap market opportunities, enhance their competitive advantage and obtain innovation returns [16]. Yu believes that with the expansion of the scope of innovation and its resources, a company

will continue to accumulate and expand. Rich innovation resources further guarantee the scale of innovation application and entrepreneurial activity. This forms a virtuous cycle that allows self-growth and environmental adaptability after open-source innovation reaches critical conditions [14,17].

Moreover, the short-sighted behavior of enterprises seeking to maximize their interests also makes it difficult to promote open-source innovation. In his 1968 account of the tragedy of the commons, Hardin wrote that users of the commons might maximize their short-term interests at the expense of their long-term interests. The actions of the minority may harm the entire group [18]. How to prevent the open-source innovation community from experiencing the tragedy of the commons with participants spontaneously innovating and sharing the innovation results are all urgent problems to be solved. Qi and Zhang pointed out that the concept of "peers" is one of the core values of the open-source community. Knowledge-sharers expect that their requirements will be satisfied by other sharers when they need access to knowledge in the future. However, there are many free-riders in the open-source community. They share in the benefits of public goods but rarely contribute anything to the community [19].

Therefore, choosing open-source and secondary innovation is a game played between the perceptions of interest and loss uncertainty. The premise of sharing innovation results is that the perceived benefits can compensate for the perceived loss of valuable knowledge [20]. Sengupta A and Sena V also analyzed the problem from the perspective of long-term sustainability. They pointed out that the benefits of technological and achievement improvements from open-source innovation must be balanced against the cost of higher returns from what is currently closed innovation. Different market conditions also affect the choice of open-source innovation [21]. Therefore, effectively proving that open-source innovation can bring greater benefits and stimulate the power of enterprise through open-source innovation is an important condition for the rapid promotion and uptake of an open-source innovation model.

### 2.3. The Factors Affecting the Revenues of Open-Source Innovation

There is no lack of relevant literature detailing research into the concept of open-source and open-source innovation. However, the current research mainly focuses on related concepts, such as open-source innovation motivation [22,23], open-source innovation-related intellectual property protection [24,25], etc. There is little research into revenue from open-source innovation. In the existing relevant research, the factors affecting the income from open-source innovation include knowledge absorption ability, product homogenization degree, the enterprise-strength gap, etc.

#### 2.3.1. The Capability of Absorbing Knowledge

This refers to the scenario when a particular innovation results in generating knowledge spillover, and the efficiency of other firms in learning of and utilizing the innovation results. Its strength directly determines the benefits of open-source learning enterprises and indirectly determines the loss of income experienced by open-source enterprises due to business competition. It plays an important role in affecting whether enterprises choose open-source practice as well as open-source innovation revenue. From the perspective of open-source innovation motivation, Hsu and Chang pointed out that the knowledge-sharing process is complex and difficult to predict. Open-source participants are uncertain whether their open-source behavior will trigger adverse results. This creates uncertainty about knowledge-sharing [26]. Tian and Chang also believe that other users will exhibit opportunistic behaviors that are intended to maximize their own interests and ignore the interests of others. This will deepen the uncertainty regarding knowledge-sharing [20]. From the perspective of open-source innovation revenue, IBM Analytics calculates the potential benefits of using open-source software and engaging in open-source projects. They found that investing USD 100 million to support open-source operating systems such as Linux will generate millions of dollars annually from around the world from the

knowledge sharing of open-source activities. Maintaining such an operating system costs USD 500 million a year. The open-source innovation model can effectively reduce costs and increase profits for enterprises [5]. Qi and Zhang combined open-source innovation with big data, pointing out that open-source big data cooperation assets have an outstanding value in terms of spillover effect [19]. Furthermore, Hagedoorn and Wang, Spencer, and Van Wijk combined the typical cooperation and competition model of the open innovation alliance to prove that knowledge-sharing has significant benefits, even for enterprises with competitive relationships [27–29].

### 2.3.2. The Degree of Product Homogeneity

Some scholars also express this as the intensity of competition (in fact, the two have a very close positive relationship). This refers to enterprises' competition in the marketplace, their benefits from product innovation or imitation (such as occupying a greater market share, etc.), and the resulting impact on income from other enterprises. The higher the degree of homogenization of enterprise products, the greater the intensity of competition. On the one hand, this means that innovation brings greater comparative advantage benefits and a company can seize the market share of other enterprises to a greater extent. On the other hand, this also means that the benefits garnered from learning about the innovation achievements of other enterprises are more significant. Through empirical research, Dedman, Lennox, and Darrough show that the stronger the market competition, the less willing companies are to disclose innovation results [30,31], and the less willing they are to innovate. However, according to the conclusions of Diao, Ma, and Hughes, the intensity of competition positively impacts the innovation disclosure of enterprises [32,33]. De Marco and Leckel believe that the impact of enterprise product homogenization on small, medium-sized, and large enterprises is heterogeneous [34,35].

### 2.3.3. Enterprise Strength

In recent years, more and more scholars have reported that large enterprises often have different open-source innovation choices than small and medium-sized enterprises. Competition intensity and many other factors have an impact of different strengths on enterprises that is often heterogeneous. Qi and Zhang believe that large enterprises are more inclined to use closed-source products because of business security and stability. Small and medium-sized enterprises tend to choose open-source products for cost reasons. Rammer also found that small and medium-sized enterprises are more inclined to try open-source innovation. They are eager to gain external knowledge through open-source innovation. However, due to their weak internal innovation ability and absorption ability [36] compared with strong, large enterprises, they lack the corresponding resources to coordinate and absorb the spillover value brought by open-source innovation [37]. Most large enterprises that dominate the market have stronger technological innovation ability and will face high risks when choosing to be open-source. Therefore, large enterprises choose open-source innovation more when it concerns establishing technical standards and optimizing organizational evolution strategies and innovation capabilities [8,9,38]. In contrast, small and medium-sized enterprises mainly choose open-source to reduce innovation costs and enhance innovation ability [7,39]. Hoab pointed out that policymakers need to be aware of the different types of innovators in SMEs and explore the appropriate innovative models [37].

### 2.3.4. Other Factors

In addition to the above influencing factors, secondary innovation ability, the cost, the value of the innovation results, and the number of participants will also affect the benefits garnered from open-source innovation. Secondary innovation ability refers to the degree of value increase for learning enterprises based on the source results during secondary innovation, which represents the ability of secondary innovators to modify the source results [5]. The stronger the secondary innovation ability, the greater the innovation

benefits of the learning enterprises. The cost of secondary innovation refers to the cost paid by a learning enterprise to carry out secondary innovation. It affects the income of learning enterprises, and some scholars will set it as a fixed value in their research [40]. The value of innovation results includes the source value and secondary innovation value of open-source enterprises. The source results' value is reflected in their accessibility and availability [5]. It affects the income of the innovation revenue of open-source enterprises and knowledge spillover to learning enterprises. The value of secondary innovation results affects both the innovation revenue of learning enterprises and the revenue brought by knowledge spillover to open-source enterprises. Some scholars assume the value of the source results to be 1 for the purposes of calculation [40]. Maxwell, Sengupta, and Sena proposed that the number of participants positively impacts open-source innovation revenue [5,21].

### 2.4. The Deficiencies in the Current Study

Many factors affect the revenue from open-source innovation, and many factors have a strong correlation. This is a significant difficulty in the current research. In recent years, many scholars have selected multiple factors affecting the revenue from open-source innovation for comprehensive analysis. For example, Wang and Gao analyzed the competition, spillover, and diffusion effects as model variables [40]. Diao and Ma explored the comprehensive impact of network externalities, competition intensity, and innovation degree on the benefits of open-source innovation [32]. Johnson took the cost of innovation and the number of companies participating in open-source activities as significant variables [10].

The following problems can be found throughout the relevant literature on open-source innovation revenue: (1) the model assumptions are too strict. Wang and Gao assume the innovation cost as a constant value, while Diao and Ma assume that firms can fully absorb knowledge and technology. (2) The model variables are not mutually independent, and there are inter-influential relationships among them. For example, the intensity of competition and the learning ability of firms are intrinsically related, and the degree of homogeneity of firms has an isotropic relationship with both of the above. (3) The model is not precise and ignores the possible non-linear relationships between open-source innovation revenues and the relevant variables; thus, the conclusions are often too generalized.

Given the shortcomings of the existing literature, this paper, based on the open-source game model of two enterprises, has considered many of the influencing factors involved in the relevant papers to make the scenario more realistic. To solve the problem of variable independence, this paper refines the relevant influencing factors into two unrelated model variables, namely, the degree of product homogeneity and the strength gap between firms. This paper analyses the influence of product homogeneity and the strength gap between firms on factors such as competition intensity. It builds a corresponding non-linear functional relationship to make an accurate model. The paper attempts to address the following two questions: firstly, what is the impact of the degree of product homogeneity on the revenue from open-source innovation? Secondly, what is the impact of the strength gap between firms on the revenue from open-source innovation? This paper aims to provide a new perspective and research framework for the study of enterprise revenues from open-source innovation and to provide policy suggestions for designing incentive mechanisms for open-source innovation.

### 3. Model Assumptions and Rationale

#### 3.1. Model Assumptions

To simplify the model, this paper only discusses open-source innovation game theory as applied to two firms, defining the open-source enterprise as Enterprise A and the learning enterprise as Enterprise B. The value of the source outcome is set as a fixed value of 1 to facilitate the calculations. Moreover, the influence of factors such as policy orientation, the intrinsic motivation of the enterprise, the cost of disclosure of the innovation outcome, and

the motivation to establish industry standards is not considered. This section is based on the theoretical analysis of competition intensity, knowledge absorption ability, secondary innovation ability, the secondary cost of innovation in terms of open-source innovation income, and the relationship with enterprise homogenization and tries to deduce the functional relationship. The model's assumptions are as follows: (1) it is assumed that firms' capabilities in various aspects such as talents, management, and the market match their strength. That is to say, the strength of firms can represent the size of their market share, competition ability, and scientific research ability, without considering situations such as the mismatch between firms' strength and scientific research ability; (2) it is assumed that the innovation revenues of Enterprises A and B are not limited by their production capability and can satisfy all the demands of consumers in the market. That is to say, the market occupied by Enterprises A and B is formed by each other's efforts at grabbing the attention of the fixed consumer market; (3) it is assumed that there is no further innovation after the secondary innovation. The innovation process is only considered up to the second innovation of learning Enterprise B. The third or more innovations are not considered.

*3.2. Parameter Setting*

This paper is based on open-source innovation game theory and supposes the value of the innovation results of the open-source Enterprise A to be 1. Other variables are set as follows: (1) let $x$, which is the degree of homogeneity, be the degree of product substitutability of the two firms. Homogeneity includes the convergence of product performance, business model, management, etc. [41]. It is hereafter expressed as product homogeneity for simplicity. If $x \in [0,1]$, $x = 0$, their products are completely heterogeneous; if $x = 1$, they are completely homogeneous. (2) Let $\alpha$, which is the competitive intensity, be the degree of the impact of innovation success on the revenue of competing firms, i.e., the revenues that change as a result of the competitive relationship/value of the original innovation result, reflecting the size of the firm's competitiveness. The competitive intensity of Enterprise A is $\alpha_1$, and the competitive intensity of Enterprise B is $\alpha_2$, $\alpha \in [0,1]$. (3) Let $\gamma$, which is the capability of knowledge absorption, be the extent to which the enterprise can learn to use the innovation knowledge when it is fully disclosed, i.e., the value learned/the value of the learning knowledge, reflecting the strength of the firm's learning capability. The knowledge absorption capability of Enterprise A is $\gamma_1$, and the knowledge absorption capability of Enterprise B is $\gamma_2$, $\gamma \in [0,1]$. (4) Let $\mu$ ($\mu \geq 0$), which is the capability of secondary innovation, be the magnitude of the value that is added by the secondary innovation of firms, i.e., the value increment of secondary innovation/the value of the original innovation outcome. (5) Let $C$, which is the cost of secondary innovation, be the expected cost of secondary innovation/the value of the original innovation outcome. As the first three variables all denote the change in revenues compared with the value of the innovation outcomes, to make the calculation easy, the cost of secondary innovation is also mentioned here compared with the magnitude of the value of the original innovation result. Meanwhile, since innovation costs include both innovation input and innovation success probability, in order to simplify the presentation, the innovation success probability is also included in the innovation input in this paper. That is to say, the cost of secondary innovation mentioned in this paper actually refers to the expected cost of secondary innovation (innovation input/innovation success probability) compared with the value of the original innovation outcome, which will not be stated hereinafter. (6) Let $m$, which is the strength gap between firms, be Enterprise B's strength/Enterprise A's strength. Since the competitiveness and learning ability of the two firms in the hypothesis are proportionate to the enterprise strength, it can be concluded that $\frac{\alpha_2}{\alpha_1} = \frac{\gamma_2}{\gamma_1} = m$.

The meanings and definition domain of values of all the variables are shown in Table 1.

**Table 1.** Variables and parameter table.

| Variable | Meaning | Definition Domain |
|---|---|---|
| $x$ | Degree of homogeneity between Enterprise A and Enterprise B products (alternative degree). | $x \in [0, 1]$ |
| $\alpha_1$ | The competition intensity of Enterprise A, that is, the degree of change of Enterprise B's income caused by the change in per-unit income of Enterprise A. | $\alpha_1 \in [0, 1]$ |
| $\alpha_2$ | The competition intensity of Enterprise B, that is, the degree of change of Enterprise A's income caused by the change in per-unit income of Enterprise B. | $\alpha_2 \in [0, 1]$ |
| $\gamma_1$ | Under the condition of complete disclosure of innovation knowledge, Enterprise A can learn to use the degree of Enterprise B's secondary innovation results. | $\gamma_1 \in [0, 1]$ |
| $\gamma_2$ | Under the condition of complete disclosure of innovation knowledge, Enterprise B can learn to use the degree of Enterprise A's secondary innovation results. | $\gamma_2 \in [0, 1]$ |
| $\mu$ | The increased value range of Enterprise B for secondary innovation. | $\mu \geq 0$ |
| $C$ | Relative expected cost of Enterprise B's secondary innovation, namely, the average single input of secondary innovation (success rate of secondary innovation $\times$ source innovation achievement value). | $C > 0$ |
| $m$ | Enterprise B's strength is the multiple of Enterprise A's strength. | $m > 0$ |

### 3.3. Function Setting and Rationale

### 3.3.1. Competition Intensity

Most of the current studies on the factors influencing the revenue from open-source innovation involve competition intensity. As mentioned above, Dedman, Lennox, Darrough, Li Qingman and Diao Lilin, Ma Yanan, Hughes, etc., have drawn completely different conclusions regarding the impact of competitive intensity on the revenue of enterprise open-source innovation [42]. This paper argues that a major reason for the differential findings is that most studies equate competitive intensity with the degree of product homogeneity, ignoring their non-linear relationship. The higher the degree of homogeneity $x$ is, the greater the competition intensity will be, showing a significant positive relationship. However, the homogeneous change between $x$ and $\alpha$ does not represent a linear relationship, as illustrated here in terms of consumer preferences.

Competition intensity is essentially a function of the ability to change the opposition's customers' product choice preferences and capture their users. The higher the degree of homogeneity is, the more substitutable the two firms' products will be to consumers, and the greater the intensity of competition. Therefore, competition intensity is positively related to product homogeneity. However, when the degree of product homogeneity varies, the incremental change in competition intensity caused by the change in product homogeneity also varies, and their relationship is not linear. When the degree of product homogeneity between two firms is very low and $x \to 0$, products tend to be irrelevant, and the consumers are very different in their demands. Hence, it is difficult for consumers to change over to another firm's product due to its innovation. At this point, the change in product homogeneity has little effect on competition intensity $\frac{\Delta \alpha}{\Delta x}$; when the degree of homogeneity is very high, $x \to 1$, and the products tend to offer the same functions, there is nearly no difference in the satisfaction degree of consumers.

However, consumers have a certain degree of loyalty to the product due to user viscosity. When product differences are minimal, there is no reason for consumers to change their consumption habits because of small changes in the product, so the impact of changes in product homogeneity on competition intensity $\frac{\Delta \alpha}{\Delta x}$ is also minimal. Consumers who are likely to change their consumption preferences due to minor changes in products are those who have difficulty in choosing products. They are in the dilemma of choosing between two different products with their advantages and disadvantages, i.e., the "middle ground" of the consumer market. Small changes in the product are likely to change their choice.Consumers do not clearly prefer a product that is too different in terms of the

needs it meets, nor do they maintain their consumption habits if the differences are too small. At that point, changes in homogeneity have a greater impact on the competition intensity $\frac{\Delta \alpha}{\Delta x}$. In summary, although homogeneity and competition intensity are bound to change in the same direction, the impact of homogeneity on the competition intensity $\frac{\Delta \alpha}{\Delta x}$ should be larger first and then smaller; that is, the rate of change $\frac{d\alpha}{dx}$ is approximately a quadratic function with a downward opening. Therefore, $\alpha$ is a cubic function of $x$. Let $\alpha(x) = mx^3 + nx^2 + px + q$ and let $m, n, p, q$ be constants. If $x = 0$, $\frac{d\alpha}{dx} = 0$, $\alpha(x) = 0$; if $x = 1$, $\frac{d\alpha}{dx} = 0$, $\alpha(x) = 1$. Therefore, $\alpha(x) = -2x^3 + 3x^2$.

In addition to the degree of homogenization, the strength gap between firms affects the competition intensity. According to Section 3.1 (1), the competitive intensity and strength of Enterprise A and Enterprise B are proportional. From $\frac{\alpha_2}{\alpha_1} = m$, we can conclude that:

$$\alpha_1(x) = -2x^3 + 3x^2, \alpha_2(x, m) = -2mx^3 + 3mx^2 (0 \le x \le 1, m > 0).$$

### 3.3.2. Absorption Knowledge Capability

The high degree of knowledge spillover from open-source innovation allows firms to acquire new ideas and technologies by learning from other innovations and enhancing their gains. The size of the benefits gained from learning innovations depends on two factors: first, whether the innovation achievements can be learned easily; second, the strength of the firm's learning ability. Whether the innovation results are easy to learn is related to the degree of enterprise homogeneity. The higher the homogeneity is, the more similar the two firms will be in terms of products, management, and marketing, making them easier to learn from. Conversely, if the degree of homogeneity is lower, the differences in products will be greater. Due to the lack of similar experiences, learning will be more difficult, and the ability to absorb knowledge will be weaker. The learning ability of an enterprise is reflected in the strength of its scientific and technological talents. According to Section 3.1 (1), the stronger an enterprise is, the more scientific and technological talents it will have, and the more it will learn from other firms' innovations, i.e., the stronger its capability of absorbing knowledge will be. Thus, an enterprise's capability of absorbing knowledge is determined by both the degree of product homogeneity and the strength of the enterprise.

Although the degree of homogeneity $x$ and knowledge absorption capability $\gamma$ change in the same direction, there are differences in the speed of change in terms of absorption capability $\frac{\Delta \gamma}{\Delta x}$. When the degree of homogeneity is high, the innovation patterns are similar. Learning knowledge is not only easy but also fast; when the degree of homogeneity is low, the innovation patterns differ greatly. The learning hindrance formed by product heterogeneity is strong, and the efficiency of knowledge absorption is low. Hence, the change rate of knowledge absorption capability regarding the degree of homogeneity $\frac{d\gamma}{dx}$ increases continuously and the two show a concave function relationship. When the product is completely heterogeneous, it is almost impossible to learn; when it is completely homogeneous, the enterprise can learn completely without hindrance. Therefore, if $x \to 0$, $\gamma \to 0$; if $x \to 1$, $\gamma \to 1$. Let $\gamma(x)$ be a simple quadratic function about $x$, $\gamma(x) = ax^2 + b$ ($a, b$ as constants). Let us enter $x = 0$, $\gamma = 0$; $x = 1$, $\gamma = 1$, and the solution is $\gamma(x) = x^2$. Since the spillover effect of Enterprise A and Enterprise B is proportional to strength, it can be established that $\gamma_1(x) = x^2$, $\gamma_2(x, m) = mx^2$ ($0 \le x \le 1, m > 0$).

### 3.3.3. Secondary Innovation Capability

As mentioned in Section 3.1, open-source innovation can also obtain revenue from the learning of innovation results and secondary innovation compared to traditional closed innovation. For example, in open-source software, secondary innovation is carried out through the optimization and improvement of the source code to deepen the value of results. The strength of secondary innovation capability and the increase in the value of source results are also important factors affecting the revenue from open-source innovation. For one thing, the stronger the relative strength of the enterprise is, the higher the intensity of scientific and technological talents will be; also, the more able the enterprise will be to

carry out value extension and in-depth optimization, based on the source results, and the stronger the secondary innovation ability will be. For another thing, the optimization and improvement of secondary innovation lie in the heterogeneous part of the enterprise. That is to say, the improvement is presented in terms of its advantages, but the source results do not have them, and a new value is added. Therefore, the secondary innovation capability is positively related to the relative strength and the degree of heterogeneity of the enterprise, from which it can be concluded that:

$$\mu(x, m) = m(1 - x) \ (0 \leq x \leq 1, m > 0).$$

3.3.4. Secondary Innovation Cost

Secondary innovation cost is the expense of innovation, which comes from the cost that the heterogeneous part of the product requires the enterprise to learn and master. The greater the difference between the source results and its own innovation mode, the more time and funds it will take to master the knowledge and technology, and the higher the cost of purchasing production equipment and hiring relevant talents will be. Conversely, the weaker the heterogeneity and the higher the degree of homogeneity, the lower the cost of secondary innovation.

The secondary innovation cost $C$ originates from the cost of enterprise heterogeneity $1 - x$, which is positively correlated but is not linear. When the product is completely heterogeneous, innovation is almost impossible for unrelated goods. As the heterogeneity diminishes, more content in the source results matches the existing experience, and the difficulty of innovation will decrease faster. Similar to the ability to absorb knowledge, the rate of change of the secondary innovation cost on the degree of heterogeneity $\frac{dC}{d(1-x)}$ keeps increasing. The two show a concave functional relationship, which can be set as $C(x) = a(1 - x)^2 + b$ (with $a, b$ as constants). In the case of complete homogeneity, the enterprise can acquire all the knowledge and technology of the source results without paying any cost (but that part of heterogeneity cannot be optimized, so its innovation value is 0); in the case of complete heterogeneity, it is necessary to innovate from learning about unrelated products, and the cost of learning and mastery is equal to the cost of all the value of the result. Therefore, if $x = 1$, $C = 0$; if $x = 0$, $C = 1$. Therefore, $C = (1 - x)^2$ $(0 \leq x \leq 1)$.

## 4. Game Process and Model Solution

### 4.1. Game Process

Referring to the change process of revenues from open-source innovation in Section 3.1, the game process can be depicted as follows:

Step 1: Enterprise A produces innovative results with a value of 1. The competition effect of Enterprise A on Enterprise B reduces the revenues of Enterprise B. At this point, the revenue of Enterprise A is 1; the revenue of Enterprise B is $-\alpha_1$.

Step 2: Enterprise A chooses open-source practice and fully discloses its knowledge achievements, and Enterprise B increases revenues at $\gamma_2$ by learning from Enterprise A's innovation achievements; meanwhile, Enterprise B increases its revenues, which causes the revenues of Enterprise A to decrease at $\alpha_2\gamma_2$ due to the competition effect. At this point, the revenue of Enterprise A is $1 - \alpha_2\gamma_2$; the revenue of Enterprise B is $\gamma_2 - \alpha_1$.

Step 3: Enterprise B decides whether or not to conduct secondary innovation. If they do not, the game ends, and the final revenues of the two firms are thus: the revenue of Enterprise A is $1 - \alpha_2\gamma_2$; the revenue of Enterprise B is $\gamma_2 - \alpha_1$. If Enterprise B decides to make the second innovation, Enterprise B will pay the innovation cost $C$ and gain the revenue $\mu$. Meanwhile, due to the competition effect, Enterprise A reduces its revenue by $\alpha_2\mu$, and Enterprise B must disclose its second innovation results (which are represented by the value $\mu$). At this point, the revenue of Enterprise A is $1 - \alpha_2\gamma_2 - \alpha_2\mu$; the revenue of Enterprise B is $\gamma_2 - \alpha_1 - C + \mu$.

Step 4: Enterprise A learns from the secondary innovation results disclosed by Enterprise B and increases its revenue by $\gamma_1\mu$, which brings the revenue reduction of Enterprise B to $\alpha_1\gamma_1\mu$ because of the competition effect. The final revenue of Enterprise A is $1 - \alpha_2\gamma_2 - \alpha_2\mu + \gamma_1\mu$; the revenue of Enterprise B is $\gamma_2 - \alpha_1 - C + \mu - \alpha_1\gamma_1\mu$.

The competitive intensity, knowledge absorption ability, secondary innovation capability, and secondary innovation cost, when expressed by product homogeneity and the enterprise-strength gap, are substituted into the expression of enterprise revenue in the four stages of open-source innovation activities. If we suppose that the revenues of Enterprise A in the four stages are $\{y_1, y_2, y_3, y_4\}$, respectively, those of Enterprise B are $\{g_1, g_2, g_3, g_4\}$, respectively, and the total revenues of the two firms are $\{f_1, f_2, f_3, f_4\}$, respectively; then, we can work out that:

$y_1 = 1$
$y_2 = 1 - m^2 x^4 (3 - 2x)$
$y_3 = 1 - m^2 x^4 (3 - 2x) - m^2 x^2 (3 - 2x)(1 - x)$
$y_4 = 1 - m^2 x^4 (3 - 2x) - m^2 x^2 (3 - 2x)(1 - x) + mx^2(1 - x)$
$g_1 = 2x^3 - 3x^2$
$g_2 = mx^2 + 2x^3 - 3x^2$
$g_3 = mx^2 + 2x^3 - 3x^2 - (1 - x)^2 + m(1 - x)$
$g_4 = mx^2 + 2x^3 - 3x^2 - (1 - x)^2 + m(1 - x) - mx^4(3 - 2x)(1 - x)$
$f_1 = 1 + 2x^3 - 3x^2$
$f_2 = 1 - m^2 x^4 (3 - 2x) + mx^2 + 2x^3 - 3x^2$
$f_3 = 1 - m^2 x^4 (3 - 2x) - m^2 x^2 (3 - 2x)(1 - x) + mx^2 + 2x^3 - 3x^2 - (1 - x)^2 + m(1 - x)$
$f_4 = 1 - m^2 x^4 (3 - 2x) - m^2 x^2 (3 - 2x)(1 - x) + mx^2(1 - x) + mx^2 + 2x^3 - 3x^2$
$- (1 - x)^2 + m(1 - x) - mx^4(3 - 2x)(1 - x)$

*4.2. Data Calibration*

It can be seen from the functional relationship that open-source innovation revenue is a binary function of the gap between the product homogenization degree and the relative strength of enterprises. Below, we will analyze the impact of the product homogenization degree and enterprise relative strength gap on the total revenue of open-source enterprises, learning enterprises, and open-source innovation activities through the model solution. Finally, the conditions for the second innovation of learning enterprises are obtained.

4.2.1. The Impact of Product Homogeneity and the Relative Strength of the Enterprise on the Revenue of Enterprise A

**Hypothesis (H1).** *When the strength of Enterprise A does not far exceed that of Enterprise B, the enhancement of the relative strength of Enterprise B will reduce the revenue of Enterprise A in open-source innovation activities.*

**Proof.**
The partial derivative of the equation $y_4$ with respect to the degree of homogeneity $x$ can be obtained as follows:

$$\frac{\partial y_4}{\partial x} = mx \left[ 2 - 3x - m \left( -10x^3 + 20x^2 - 15x + 6 \right) \right]$$

We draw $y = -10x^3 + 20x^2 - 15x + 6$ as is shown in Figure 1.

It can be ascertained that $-10x^3 + 20x^2 - 15x + 6 > 0$ is permanently established. Hence, the condition of $\frac{\partial y_4}{\partial x} > 0$ is $m < \frac{2 - 3x}{-10x^3 + 20x^2 - 15x + 6}$, and the image of $y = \frac{2 - 3x}{-10x^3 + 20x^2 - 15x + 6}$, as is shown in Figure 2.

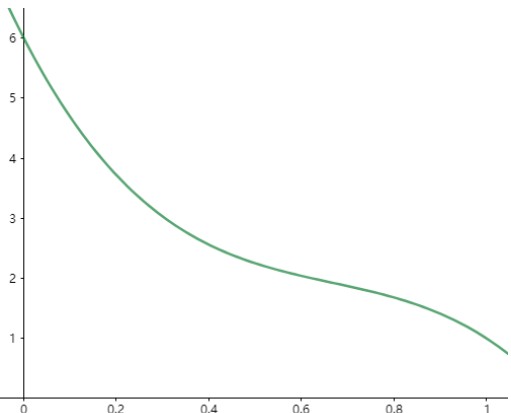

**Figure 1.** Function graph of $y = -10x^3 + 20x^2 - 15x + 6$.

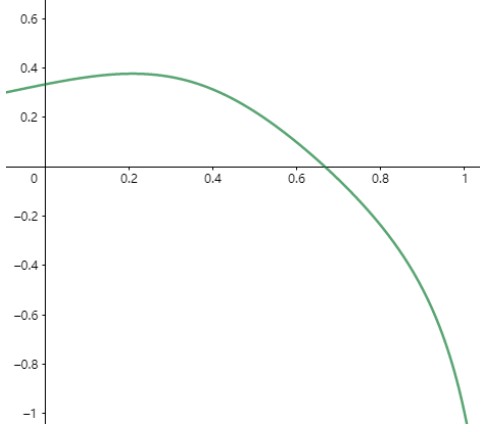

**Figure 2.** Function graph of $y = \frac{2-3x}{-10x^3+20x^2-15x+6}$.

This can be solved approximately if $x > 0.66$, $\frac{2-3x}{-10x^3+20x^2-15x+6} < 0$ is permanently established, and $\frac{2-3x}{-10x^3+20x^2-15x+6} < 0.38$ is also constantly established. Therefore, if $x > 0.66$ or $m > 0.38$, the revenue of open-source Enterprise A decreases with the increasing degree of homogeneity; if $x < 0.66$ and $m < \frac{2-3x}{-10x^3+20x^2-15x+6}$, the revenue of Enterprise A increases with the increasing degree of homogeneity. $\square$

**Hypothesis (H2).** *When the strength of Enterprise A does not far exceed that of Enterprise B, the enhancement of the relative strength of Enterprise B will reduce the revenue of Enterprise A in open-source innovation activities.*

**Proof.**

The partial derivative of the equation $y_4$ for the relative strength gap of the enterprise $m$ can be obtained as $\frac{\partial y_4}{\partial m} = x^2\left[1 - x - 2m\left(-2x^3 + 5x^2 - 5x + 3\right)\right]$.

We draw $y = -2x^3 + 5x^2 - 5x + 3$ as is shown in Figure 3.

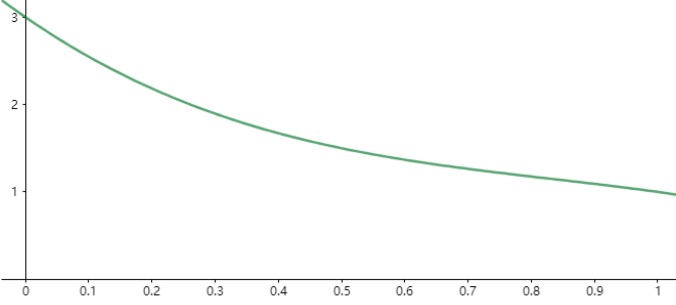

**Figure 3.** Function graph of $y = -2x^3 + 5x^2 - 5x + 3$.

It can be concluded that $-2x^3 + 5x^2 - 5x + 3 > 0$ is permanently established.

Hence, the condition of $\frac{\partial y_4}{\partial m} > 0$ is $m < \frac{1-x}{2(-2x^3+5x^2-5x+3)}$, and the image of $y = \frac{1-x}{2(-2x^3+5x^2-5x+3)}$ as is shown in Figure 4.

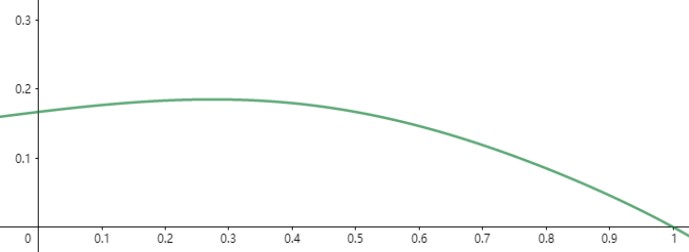

**Figure 4.** Function graph of $y = \frac{1-x}{2(-2x^3+5x^2-5x+3)}$.

The approximate solution can be obtained as follows: if $m > 0.19$, $\frac{\partial y_4}{\partial m} < 0$; if $m < \frac{1-x}{2(-2x^3+5x^2-5x+3)}$, the revenue of open-source Enterprise A increases with the improvement of Enterprise B's strength relative to itself. □

### 4.2.2. The Impact of Product Homogeneity and Enterprise Relative Strength Gap on Enterprise B's Revenue

**Hypothesis (H3).** *When the degree of homogeneity is medium, the further increase of the degree of homogeneity will reduce the revenue from open-source innovation of the learning-oriented Enterprise B; when the degree of homogeneity is high, and the relative strength of Enterprise B is strong, the further increase of the degree of homogeneity will raise the revenue from open-source innovation of learning-oriented Enterprise B.*

**Proof.**

The partial derivative of the equation $g_4$ concerning the degree of homogeneity $x$ can be obtained as follows: $\frac{\partial g_4}{\partial x} = 6x^2 - 8x + 2 - m(12x^5 - 25x^4 + 12x^3 - 2x + 1)$.

We draw $y = 6x^2 - 8x + 2$ and $y = 12x^5 - 25x^4 + 12x^3 - 2x + 1$ as are shown in Figures 5 and 6.

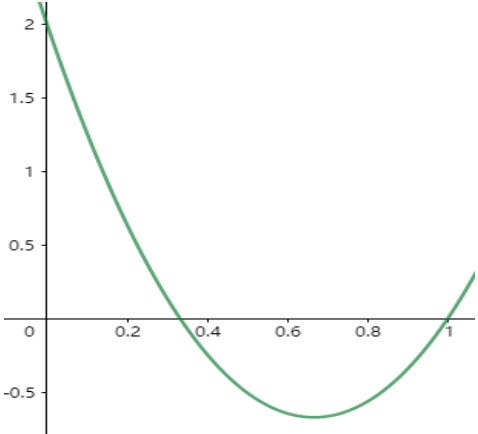

**Figure 5.** Function graph of $y = 6x^2 - 8x + 2$.

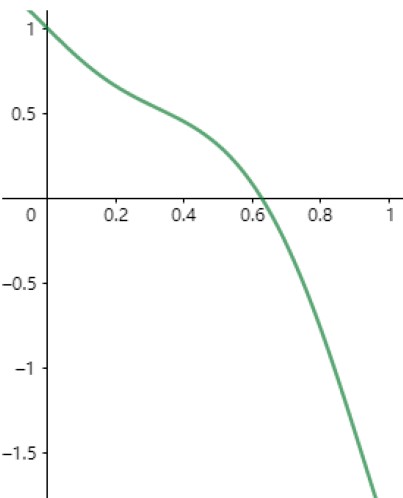

**Figure 6.** Function graph of $y = 12x^5 - 25x^4 + 12x^3 - 2x + 1$.

The approximate solution can be obtained as follows:

If $x < 0.33$, $m < \frac{6x^2 - 8x + 2}{12x^5 - 25x^4 + 12x^3 - 2x + 1}$ or if $x > 0.63$ and $m > \frac{6x^2 - 8x + 2}{12x^5 - 25x^4 + 12x^3 - 2x + 1}$, $\frac{\partial g_4}{\partial x} > 0$. If $0.33 < x < 0.63$, then $\frac{\partial g_4}{\partial x} < 0$ is permanent. $\square$

**Hypothesis (H4).** *The stronger the relative strength of the learning-oriented Enterprise B, the greater the revenue in the process of open-source innovation.*

**Proof.**

The partial derivative of the equation $g_4$ for the relative strength gap of the enterprise $m$ can be obtained as follows: $\frac{\partial g_4}{\partial m} = -2x^6 + 5x^5 - 3x^4 + x^2 - x + 1$, and the image of it as is shown in Figure 7.

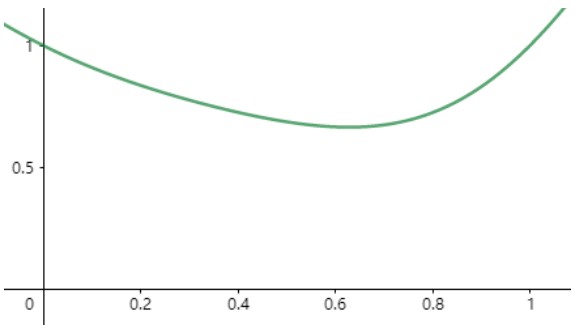

**Figure 7.** Function graph of $y = -2x^6 + 5x^5 - 3x^4 + x^2 - x + 1$.

It is shown that $\frac{\partial g_4}{\partial m} > 0$ is permanently true. $\square$

### 4.2.3. The Impact of Product Homogeneity and the Relative Strength Gap on the Total Revenue of the Two Firms

**Hypothesis (H5).** *When the strength of Enterprise B is significantly weaker than that of Enterprise A, the enhancement of Enterprise B's strength is conducive to improving the total revenue of the two firms. However, when the degree of homogeneity is relatively high and the strength of Enterprise B surpasses that of Enterprise A, the enhancement of Enterprise B's strength will reduce the total revenue of the two firms. Therefore, when the degree of homogeneity is relatively high, the more similar the strength of the two firms and the greater the total revenue from open-source innovation.*

**Proof.**

The partial derivative of the equation $f_4$ concerning the degree of homogeneity $x$ can be obtained as follows:

$$\frac{\partial f_4}{\partial x} = mx\left[2 - 3x - m\left(-10x^3 + 20x^2 - 15x + 6\right)\right] + 6x^2 - 8x + 2 - m\left(12x^5 - 25x^4 + 12x^3 - 2x + 1\right)$$

It can be concluded that the value of the function decreases as $m$ increases. □

**Hypothesis (H6).** *If the strength of Enterprise A does not greatly exceed that of Enterprise B, the improvement in the degree of homogeneity will reduce the total revenue from open-source innovation of the two firms.*

**Proof.**

The partial derivative of the equation $f_4$ for the relative strength gap of the enterprise $m$ can be obtained as follows:

$$\frac{\partial f_4}{\partial m} = -2x^6 + 5x^5 - 3x^4 - x^3 + 2x^2 - x + 1 - 2m\left(-2x^5 + 5x^4 - 5x^3 + 3x^2\right)$$

We draw $y = -2x^5 + 5x^4 - 5x^3 + 3x^2$ as is shown in Figure 8.

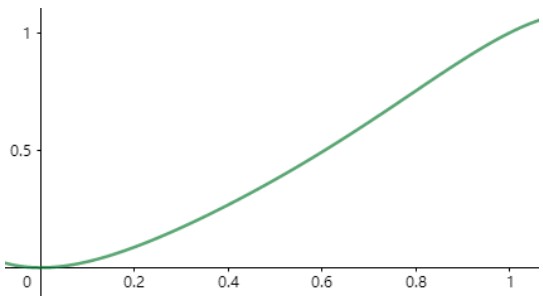

**Figure 8.** Function graph of $y = -2x^5 + 5x^4 - 5x^3 + 3x^2$.

If $m < \frac{-2x^6 + 5x^5 - 3x^4 - x^3 + 2x^2 - x + 1}{2\left(-2x^5 + 5x^4 - 5x^3 + 3x^2\right)}$, then $\frac{\partial f}{\partial m} > 0$.

The image of $y = \frac{-2x^6 + 5x^5 - 3x^4 - x^3 + 2x^2 - x + 1}{2\left(-2x^5 + 5x^4 - 5x^3 + 3x^2\right)}$ as is shown in Figure 9.

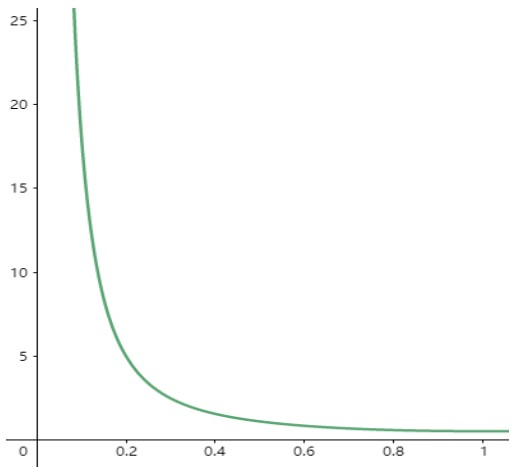

**Figure 9.** Function graph of $y = \frac{-2x^6 + 5x^5 - 3x^4 - x^3 + 2x^2 - x + 1}{2\left(-2x^5 + 5x^4 - 5x^3 + 3x^2\right)}$.

If $m < 0.5$, $\frac{\partial f}{\partial m} > 0$ is permanently established; if $x > 0.5$ and $m > 1$, then $\frac{\partial f}{\partial m} < 0$. □

### 4.2.4. Conditions for Secondary Innovation in the Learning-Oriented Enterprise B

**Hypothesis (H7).** *When the strength of Enterprise B is weaker than that of Enterprise A, Enterprise B is not willing to try a second innovation after learning the open-source results. When the degree of homogeneity is relatively high, and the strength of Enterprise B is relatively strong, Enterprise B has the motivation to conduct a secondary innovation.*

**Proof.**

If Enterprise B is willing to plan a secondary innovation, that is because the benefits of the secondary innovation are greater than the costs. The condition is as follows:

$$g_4 - g_2 = -(1-x)^2 + m\left[1 - x - x^4(3-2x)(1-x)\right] > 0$$

The image of $y = 1 - x - x^4(3-2x)(1-x)$ as is shown in Figure 10.

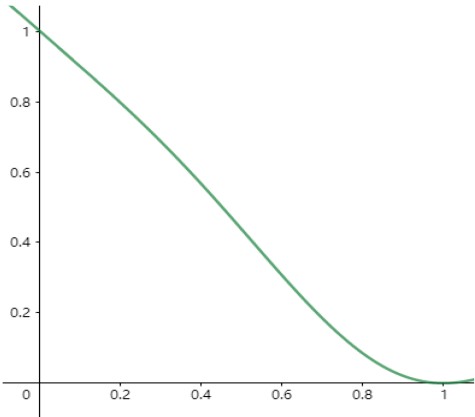

**Figure 10.** Function graph of $y = 1 - x - x^4(3-2x)(1-x)$.

Therefore, the conditions for Enterprise B to carry out secondary innovation are $m > \frac{(1-x)^2}{1-x-x^4(3-2x)(1-x)}$.

We draw $y = \frac{(1-x)^2}{1-x-x^4(3-2x)(1-x)}$ as is shown in Figure 11.

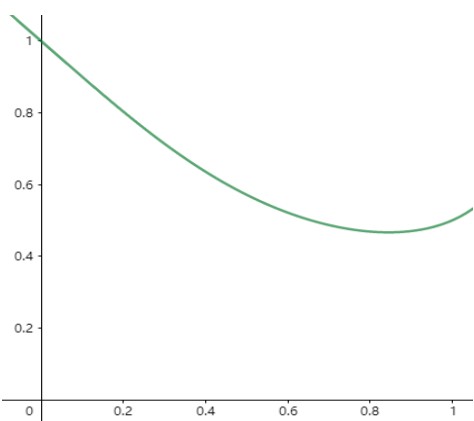

**Figure 11.** Function graph of $y = \frac{(1-x)^2}{1-x-x^4(3-2x)(1-x)}$.

The approximate solution shows that if $m < 0.46$, $g_4 < g_2$ is permanently true. □

### 5. Matlab Simulation

It is difficult to quantify the data via knowledge absorption ability, competition strength, secondary innovation ability, and innovation cost. An empirical analysis is even more difficult. To intuitively see the impact of the degree of product homogenization and the gap of enterprise strength on the enterprise income, Matlab can be used to draw three-dimensional simulation images of enterprise revenue, so as to analyze the relationship between the three.

Open-source enterprise revenue is about the function of the product homogenization degree, and the enterprise-strength gap is expressed as:

$$y_4 = 1 - m^2 x^4 (3 - 2x) - m^2 x^2 (3 - 2x)(1 - x)(1 - x) + mx^2 (1 - x).$$

The simulation image is shown in Figure 12.

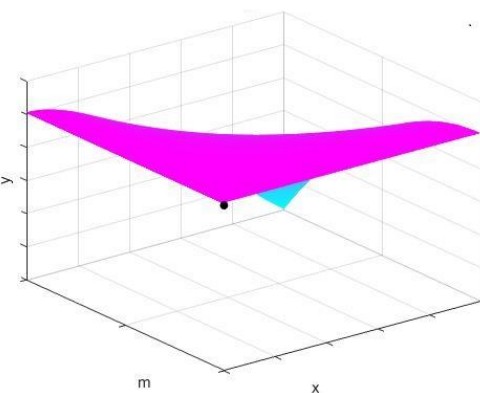

**Figure 12.** Open-source enterprise revenue as a function of the degree of product homogeneity and the enterprise power gap.

The black point represents the zero point of the enterprise income. It can be seen that with the improvement of product homogenization and the relative strength of learning enterprises, the revenue of open-source enterprises gradually declines, which is in line with the hypothesis.

The function of learning enterprise income, between the degree of product homogenization and enterprise strength, is expressed as:

$$g_4 = mx^2 + 2x^3 - 3x^2 - (1 - x)^2 + m(1 - x) - mx^4 (3 - 2x)(1 - x).$$

The simulation image is shown in Figure 13.

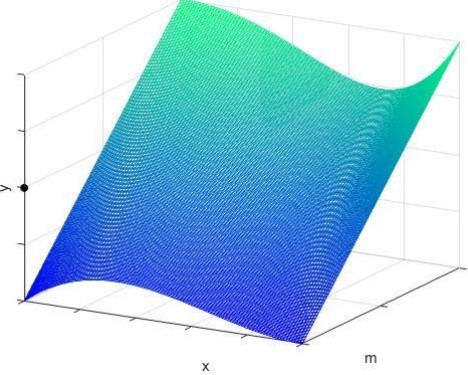

**Figure 13.** The function graph of learning enterprise income on product homogeneity degree and enterprise-strength gap. Note: The lighter the color, the higher the learning enterprise income.

It can be seen that the representation of learning enterprise revenue is a complex, wavy image. In general, the strength improvement of learning enterprises can improve the overall income. However, the impact of the change of homogenization degree on enterprise income fluctuates. The peak of the "wave" is reached at moderate homogeneity, but with the increase in homogenization, corporate earnings will then enter the trough. When the learning enterprise is relatively strong, it will be located at the upper right corner of the "wave", and improve the income again. This also explains why the relationship between learning enterprise revenue and product homogenization degree is extremely complex and is in line with the hypothesis.

The total market return as a function of the degree of product homogenization and the gap in enterprise-strength is expressed as:

$$f_4 = 1 - m^2 x^4 (3 - 2x) - m^2 x^2 (3 - 2x)(1 - x) + mx^2 (1 - x) + mx^2 + 2x^3 - 3x^2$$
$$-(1 - x)^2 + m(1 - x) - mx^4 (3 - 2x)(1 - x).$$

The simulation image is shown in Figure 14.

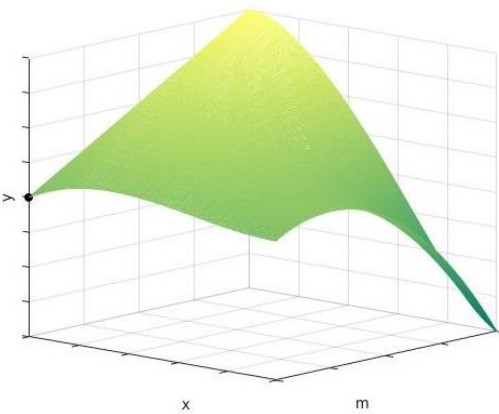

**Figure 14.** The function graph of total market return on the degree of product homogeneity and the enterprise power gap. Note: The lighter the color, the higher the total market return.

The highest point of the image is taken when the enterprise-strength gap is moderate. By assuming the specific value, it can be verified that the specific process is not repeated at the highest point. At this point, the degree of homogenization is in the middle-right position, which agrees with the hypothesis.

## 6. Research Conclusions and Policy Recommendations for Open-Source Community Construction

It can be seen from the conclusions of the model that the overall income of open-source innovation has a non-linear relationship with the degree of enterprise homogenization and the gap in enterprise strength. Its changing trend is more complex, which also explains the conclusions of different scholars on the impact of the degree of homogeneity and the income of open-source innovation. However, in general, high homogeneity and a large strength gap are not conducive to open-source innovation. A moderate degree of homogenization and similar enterprise strength are conducive to improving the total revenue from open-source innovation. High homogeneity makes the competition effect cause excessive revenue loss for open-source firms, resulting in the decline of total market revenue. A low threshold of imitation and difficulties of secondary innovation make open-source activities less profitable. A larger strength gap is unfavorable to open-source innovation activities. Seemingly, a strong enterprise can gain more benefits. However, the disadvantageous position of the weak enterprises in the process of innovation achievement, sharing in the learning absorption and competition effect, can lead to a reduction in the total revenue from open-source innovation activities. In addition, weak firms will be forced to withdraw from open-source innovation activities. This will be due to their relative

disadvantages or their choice of "free-riding" instead of conducting independent secondary innovation after the strong firms publish their innovation results. This will also make the strong firms unable to obtain ideal results and profits. Therefore, all enterprises must seek other enterprises with moderate homogenization as co-participants in open-source activities.

The marginal contribution of this paper lies in the following three parts: (1) by constructing the non-linear relationship between competitive intensity, knowledge absorption capacity, secondary innovation capacity, secondary innovation cost and other factors and the enterprise's product homogeneity degree, the enterprise's strength gap. The benefit of open-source innovation can be expressed more intuitively by means of binary function. This suggests the conclusion that a moderate degree of homogeneity is beneficial to the improvement of open-source innovation income. (2) By analyzing the heterogeneity of the influence of competitive intensity and other factors on different firms, the differences in previous research conclusions can be better explained. (3) The conditions of secondary innovation are clarified to provide a reference for enterprise open-source innovation partners to choose secondary innovation options.

More specifically, to ensure the sustainability of open-source innovation, the following suggestions are put forward regarding the construction of an open-source community.

The establishment of open-source communities should attract companies with similar strengths as much as possible; they should establish open-source community cooperation at different levels according to the strength of firms as much as possible, and avoid the large gap between the selected open-source communities or the participants and their strength. For some small and medium-sized firms that are small in scale, weak in strength, and that find it difficult to carry out innovation and development, the government should strengthen their cooperation through guidance and encourage mergers to enhance the strength of such firms. They can first seek cooperation and communication with other firms with similar products of a similar size to form a homogeneous and competitive SME alliance. Small and medium-sized firms can cooperate in research and development, production and sales, or via a series of links. In this way, they can achieve internal economies of scale [43]. After the small firms become an enterprise alliance with certain innovation ability, they will look for suitable participants in open-source activities using the overall strength of the alliance.

While ensuring that the strength of internal firms is roughly the same, the open-source community should avoid the high degree of homogeneity of firms within the community and strive to broaden the scope of open-source innovation activities. It is necessary to focus on a product's differences in terms of electronics, new materials, and other fields. Presently, research on open-source innovation mainly focuses on open-source software. However, due to the high degree of homogeneity in software, the obvious impact of the competition effect, and the great loss of benefit of open-source firms, the advantages of sharing and the mutual development of open-source innovation cannot be given full play. In addition, high homogeneity greatly reduces the obstacles of learning imitation and increases the difficulty of secondary innovation. Moreover, the disclosure of secondary innovation results will reduce the further benefits, making it difficult for learning firms to effectively guarantee the motivation of secondary innovation and make their secondary innovation results public. Therefore, the government should pay more attention to fields such as electronics and new materials with differentiated products, make full use of the inspiration brought by differentiated knowledge and technology, break the stereotype of thinking mode, encourage the sharing of open-source innovation results in such fields, and improve the innovation benefits.

Diverse means should be adopted to enhance the sustainability of the open-source innovation model and enhance the participants' sense of belonging to the open-source community. In the long run, in open-source innovation activities, the first enterprises to attempt open-source have large revenue losses and bear greater risks. Only continuous and multi-party open-source activities can improve the revenue of the participants and ensure the sustainability of the open-source innovation model.

First, it is necessary to optimize the open-source community management model and establish an effective incentive mechanism. Continuous engagement is the most critical factor in the sustainability of the open-source community. Apache, Ubuntu, and other communities have taken relevant measures to encourage people to participate in community construction. The Apache community builds web pages to record a list of contributors and information for each Apache project. The Ubuntu Community Committee grants contributors privileges such as exclusive email suffixes, developer member titles, and other advantages [44]. These measures have effectively increased the enthusiasm of developers to participate. Through the establishment of an open-source achievement database and an information system for open-source activity participants, innovation value can also be enhanced and the sustainable relationship between enterprises and partners can be consolidated [45]. In addition, it can also stimulate the community's vitality.

Secondly, we need to strengthen the training of open-source and innovative talents. Dedicated open-source talents are the driving force for the sustainable development of the open-source community. Colleges and universities are the most important bases for cultivating talent. Encouraging students to participate in maker space and open-source projects is conducive to the promotion of the open-source model.

Finally, in order to solve the problem that the first open-source enterprise bears more risks, material subsidies and spiritual incentives should be combined. Materially, the government should give certain subsidies and policy support to open-source innovation enterprises. Spiritually, we need to strengthen the bond of mutual trust between open-source enterprises.

Nowadays, open-source innovation is rarely included in the government's innovation measures [46]. Firms that take the lead in open-source have greater benefit losses and need to bear considerable risk. For open-source innovation participants with high knowledge and a technology spillover effect, innovation willingness is closely related to whether the spillover benefits can be moderately compensated [47]. When the open-source model is not formed and the problem of the "free ride" is common, the government should give support to open-source enterprises, such as offering tax subsidies. In this way, enterprises' open-source innovation power can be stimulated [48–50]. Trust is an important factor for enterprises to build a good cooperative relationship and sustainable development [51]. The spontaneous resistance to hitchhiking behavior and the enthusiasm for secondary innovation both depend on the construction of a trust relationship between enterprises. Strengthening enterprise interaction in the open-source community and building a smooth talent exchange and learning channel can better cultivate this trust.

**Author Contributions:** Conceptualization, A.T. and Q.Q.; methodology, Q.Q.; software, Q.Q.; validation, D.D. and V.B.; formal analysis, Q.Q., Y.L. and D.T.; investigation, Q.Q. and D.D.; resources, Q.Q. and D.T.; data curation, Q.Q. and D.T.; writing—original draft preparation, Q.Q.; writing—review and editing, Q.Q., Y.L., D.D. and V.B.; visualization, D.T., Y.L. and V.B.; supervision, A.T. and D.T.; project administration, A.T. and D.T.; funding acquisition, A.T. All authors have read and agreed to the published version of the manuscript.

**Funding:** This research received no external funding.

**Institutional Review Board Statement:** Not applicable.

**Informed Consent Statement:** Not applicable.

**Conflicts of Interest:** The authors declare no conflict of interest.

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
