# Peer review of "Game Analysis of the Open-Source Innovation Benefits of Two Enterprises from the Perspective of Product Homogenization and the Enterprise Strength Gap"

_sustainability, doi:10.3390/su14095572_

Round 1

Reviewer 1 Report

Thank you for the opportunity to review this interesting research article. The topic is relevant and additional research is needed.

While the methods used in this article, grounding in existing literature is somewhat lacking. The introduction makes numerous claims that should be accompanied with references (e.g., popularity of open source, benefits, motivation for companies to use open source). There are claims from Yu Jiang and Din Yumin in the text that are not referenced correctly.

The text needs to be once more corrected to fix minor grammar and typing mistakes. Also the references should be marked correctly (e.g., use only the surnames of the authors, if more than three use "Surname et al. [1]"). All the figures should be named with a caption (with possible source) and they should be referred in the text somewhere.

The most glaring shortcoming of this article is the missing justification how it contributes to the aim of the journal. While the research work itself seems to be sound, the results are not reflected with sustainability dimensions. The policy suggestions should be made in context of sustainability, and existing literature on the topic should be considered. There exists a very well established body of literature on open source innovation and applications in context of sustainability, which should be considered in this article.

While the research work is sound, with the current short comings it can not be accepted for publication in this journal. However, if the authors do address the comments above, situation can be reconsidered.

Author Response

Point 1: While the methods used in this article, grounding in existing literature is somewhat lacking. The introduction makes numerous claims that should be accompanied with references (e.g., popularity of open source, benefits, motivation for companies to use open source)

Response 1: We have supplemented the specific data and literature on the popularity of open source innovation in the introduction, and clarified the motivation and benefits of open source innovation of large enterprises and small and medium-sized enterprises proposed by some scholars.

The supplementary information on the prevalence of open source innovation is as follows:

As early as 1999, Linux open source operating system has become the biggest competitor of Microsoft operating system, and has been encroaching on the market share of Microsoft Windows system. [2] Apache, the open source network software, has maintained the king's position in the market since its release in 2002, occupying more than 60% of the market of network server software in 2007. Nowadays, open source technology is widely used in more than 80% of global software sales, and open source technology supports more than 90% of scientific and technological innovation products [3]. Therefore, the importance of studying the benefits of open source innovation is becoming increasingly prominent. (Line 35-43)

The material supplemented with the benefits and objectives of companies adopting open source innovation follows:

On the one hand, large enterprises choose open source innovation to integrate resources and reduce costs so as to improve innovation capability [6,7,8]. On the other hand, they hope to obtain the first-mover advantage of technological innovation and establish technical standards [9]. In addition, Wichmann points out that it also includes strategic considerations and compatibility considerations [10]. Through open source innovation, small enterprises can make better use of external resources owned by partners, develop diversified products and enhance competitiveness [11]. (Line 51-57)

The supplementary references are as follows:

  1. Hamm, S. Open source innovation [EB/OL].http: //open- innovation- projects. org/faq/what do you mean by open source innovation,2005- 10- 06.
  2. Garcia,Juan Mateo and Steinmuller. W. Edward. The open source way of working:Anew paradigm for the division of labour in software development.University of Sussex Science & Policy Research Unit Working Paper,2003.
  3. Bauer A,Pizka M. The contribution of free software to software evolution.International Workshop on Principles of Software Evolution [J].Helsinki,Finland,2003.
  4. Josh Lerner,Jean Tirole. Some simple economics of open source.Journal of Industrial Economics,2002,50(2):197- 234.
  5. Wichmann. T. Firms' F/OSS aetivities:motivations and Policy implications [A].Free/Libre and F/OSS Software:Survey and Study,FLOSS Final Report.Berleeom Research GmbH:Intemational Institute of lnfonomies,2002.
  6. Massimo G. Colombo,Evila Piva,Cristina Rossi- Lamastra. Open innovation and within- industry diversification in small and medium enterprises:The case of open source software firms.Research Policy, 2014(43).

Point 2: There are claims from Yu Jiang and Din Yumin in the text that are not referenced correctly.

Response 2: We carefully compared Yu and Ding's original text: “Open mechanism, governance mechanism and enlightenment analysis of open source innovation under the background of deep digitization” and revised the sentence:“open source innovation enables developers to use relevant knowledge and technology without mastering relevant property rights and patent rights”to “open source license”.

(Please see Line 139-140)

Point 3: The text needs to be once more corrected to fix minor grammar and typing mistakes.

Response 3: We have invited native English speakers to scrutinize and polish the language.

Point 4: The references should be marked correctly (e.g., use only the surnames of the authors, if more than three use "Surname et al. [1]"). All the figures should be named with a caption (with possible source) and they should be referred in the text somewhere.

Response 4: We have carefully checked the format in the references and have named each figure that appears in the text.

Point 5: The most glaring shortcoming of this article is the missing justification how it contributes to the aim of the journal.The policy suggestions should be made in context of sustainability, and existing literature on the topic should be considered. There exists a very well established body of literature on open source innovation and applications in context of sustainability, which should be considered in this article.

Response 5: In order to more clearly reflect the significance of sustainability to open source innovation model, the third suggestion in the conclusion part is more specifically interpreted as adopting diversified means to enhance the sustainability of open source innovation model, enhance participants' sense of belonging to open source community, and supplement relevant references on the sustainability of open source innovation. We explained due to the specific open source innovation mode, the first open source enterprise bigger risk, so the sustainability of the open source innovation mode will directly influence the participants, and details from the following three aspects how to enhance the sustainability of open source innovation pattern:

(1) the optimization of the open source community management mode, establish an effective incentive mechanism (2) to strengthen the open source innovation personnel training (3) Give material subsidies to open source innovation enterprises and strengthen mutual trust between open source enterprises.

(Please see Line761-803)

The supplementary references are as follows:

  1. Chen D, HU Y, Ye L. Research on sustainable Development Model of Library Open Source Community. Library Science Research (陈大庆,胡燕菘,叶兰. 图书馆开源社区持续发展模式研究[J]. 图书馆学研究), 2011(8):51-55.
  2. Hosseini,E; Tajpour,M; Salamzadeh,A; Demiryurek,K & Kawamorita,H(2021), Resilience and Knowledge-Based Firms’Performance: The Mediating Role of Entrepreneurial Thinking. Journal of Entrepreneurship and Business Resilience, 4(2), 7-29.
  3. Tajpour, M., Salamzadeh, A., Salamzadeh, Y., & Braga, V. (2021). Investigating social capital, trust and commitment in family business: case of media firms. Journal of Family Business Management. https://doi.org/10.1108/JFBM-02-2021-0013.

Reviewer 2 Report

Dear author(s)

It was my pleasure to review your manuscript entitled “Game Analysis of Open Source Innovation Benefit of Two Enterprises Based on the Perspective of Product Homogenization and Enterprise strength gap” and advise you to prosper your current research project. In my view, your topic has touched on a critical issue in a fascinating context. However, there are many spaces to be improved in terms of argumentation, theoretical background, research method, and findings. I hope my below comments would help you develop your work into groundbreaking research in your domain.

The introduction should clearly illustrate (1) what we know (the key theoretical perspectives and empirical findings) and what do we not know (major, unaddressed puzzle, controversy, or paradox does the study addresses, or why it needs to be addressed and why this matters). And, (2) what will we learn from the study and how does the study fundamentally change, challenge, or advance scholars’ understanding. Much sharper problematization is required so that the introduction draws the reader into the paper. The introduction therefore needs to do a better job in setting the stage for the articulation of the theoretical contributions of the study. At the end of the introduction, we should have a clear idea of what the paper is about (i.e. its motivation, the gap in understanding that the paper is trying to address and summary of theoretical contributions).

The positioning of the paper is not entirely clear. The author is better to explain the gap in this article further. 

Insufficient transparency. The authors need to provide and explain more details on their method, including their sampling, data gathering and data analysis.

The current work needs a round of proofreading, I strongly recommend you revise the current version by having help of a native English speaker.

What are the theoretical and practical implications of your study and which limitations and possible future research emerge from it? At the moment. the chapter is that is now entitled as "Conclusion" should link back to the literature and show theoretical contributions, that exceed the conclusion that some literature was "inline" with the findings of the authors.

What scale has been used for analysis? What are the results of your research and how can it help your statistical community?

The authors need to draw substantive conclusions from their results, and suggest, develop recommendations for further research.

- Using the following references could be beneficial as these add more evidence to the literature review section:

Tajpour, M., Salamzadeh, A., Salamzadeh, Y., & Braga, V. (2021). Investigating social capital, trust and commitment in family business: case of media firms. Journal of Family Business Management. https://doi.org/10.1108/JFBM-02-2021-0013

Hosseini,E; Tajpour,M; Salamzadeh,A; Demiryurek,K & Kawamorita,H(2021), Resilience and Knowledge-Based Firms’ Performance: The Mediating Role of Entrepreneurial Thinking. Journal of Entrepreneurship and Business Resilience, 4(2), 7-29.

Best of luck with the further development of the paper.

Author Response

Point 1: The introduction should clearly illustrate what we know (the key theoretical perspectives and empirical findings) and what do we not know (major, unaddressed puzzle, controversy, or paradox does the study addresses, or why it needs to be addressed and why this matters).The introduction should clearly illustrate what will we learn from the study and how does the study fundamentally change, challenge, or advance scholars’ understanding. Much sharper problematization is required so that the introduction draws the reader into the paper. The introduction therefore needs to do a better job in setting the stage for the articulation of the theoretical contributions of the study. At the end of the introduction, we should have a clear idea of what the paper is about (i.e. its motivation, the gap in understanding that the paper is trying to address and summary of theoretical contributions).

Response 1: We have adjusted the introduction to include the following:

However, due to the many factors affecting the benefits of open source innovation, there are few studies on its quantitative analysis through mathematical model or empirical analysis.The uncertainty of revenue seriously hinders the promotion and secondary innovation of open source innovation model.It is urgent to demonstrate that open source innovation can achieve revenue growth.

This paper analyzes the revenue channels of open source innovation model and builds revenue function with the help of game model. In this paper, the influencing factors of various open source innovation revenue are expressed by two variables: enterprise product homogenization level and enterprise strength gap. In this way, the influence of enterprise homogenization level and strength gap on open source innovation revenue can be explored more clearly. The conclusion of this study can provide inspiration for the design of open source innovation incentive mechanism and relevant policy making of open source community participants.

(Please see Line 61-74)

Point 2: Insufficient transparency. The authors need to provide and explain more details on their method, including their sampling, data gathering and data analysis.

Response 2: Since the research in this paper is not an empirical analysis and involves many variables and it is difficult to quantify, this paper mainly discusses the influence of enterprise homogeneity and strength gap on open source innovation revenue at the theoretical level through model analysis, so as to provide directional suggestions for the promotion of open source innovation model and the construction of open source community.

Point 3: The current work needs a round of proofreading, I strongly recommend you revise the current version by having help of a native English speaker.

Response 3: We have invited native English speakers to scrutinize and polish the language.

Point 4: What are the theoretical and practical implications of your study and which limitations and possible future research emerge from it? At the moment. the chapter is that is now entitled as "Conclusion" should link back to the literature and show theoretical contributions, that exceed the conclusion that some literature was "inline" with the findings of the authors. (Please see Line ?)

Response 4: We rename the conclusion part to the research conclusion and policy suggestions constructed by the open source community in order to more clearly show the theoretical contribution of this paper. We add the explanation of the marginal contribution of this paper after sorting out the model conclusion, and add some references in this part. (Please see Line 717-727)

Point 5: What scale has been used for analysis? What are the results of your research and how can it help your statistical community?The authors need to draw substantive conclusions from their results, and suggest, develop recommendations for further research.

Response 5: Since this paper is not an empirical analysis, it has not been modified with a specific scale. In the conclusion part, we have explained in more detail the substantive suggestions of this paper on the establishment and promotion of open source model in the open source innovation community. Specific recommendations added include:

(3) Diversified means should be adopted to enhance the sustainability of the open source innovation model and enhance the participants' sense of belonging to the open source community. In the long run, in the open source innovation activities, the first enterprises in open source have large revenue losses and bear greater risks. Only continuous and multi-party open source activities can improve the revenue of participants and ensure the sustainability of the open source innovation model. 

First, it is necessary to optimize the open source community management model and establish an effective incentive mechanism. Continuous engagement is the most critical factor in the sustainability of the open source community. Apache, Ubuntu and other communities have taken relevant measures to encourage people to participate in community construction. The Apache community builds web pages to record a list of contributors and information for each Apache project. The Ubuntu Community Committee grants contributors privileges such as exclusive email suffixes, developer member titles and other [51]. These measures have effectively increased the enthusiasm of developers to participate. Through the establishment of open source achievement database and information system for open source activity participants, the innovation value can also be enhanced and the sustainable relationship between enterprises and partners can be consolidated[52]. In addition, it can also stimulate the community vitality. 

Secondly, we need to strengthen the training of open-source and innovative talents. Dedicated open source talents are the driving force for the sustainable development of the open source community. Colleges and universities are the important bases for cultivating talents. Encouraging students to participate in maker space and open source projects is conducive to the promotion of the open source model.

Finally, in order to solve the problem that the first open source bears more risks, material subsidies and spiritual incentives should be combined. Materially, the government should give certain subsidies and policy support to open source innovation enterprises. Spiritually, we need to strengthen the mutual trust between open source enterprises.

(Please see Line 761-789)

Trust is an important factor for enterprises to build a good cooperative relationship and the sustainable development of [58]. The spontaneous resistance to hitchhiking behavior and the enthusiasm of secondary innovation both depend on the construction of the trust relationship between enterprises. Strengthening enterprise interaction in the open source community and building a smooth talent exchange and learning channel can better cultivate this trust.

(Please see Line 798-803)

Point 6: Using the following references could be beneficial as these add more evidence to the literature review section:

Tajpour, M., Salamzadeh, A., Salamzadeh, Y., & Braga, V. (2021). Investigating social capital, trust and commitment in family business: case of media firms. Journal of Family Business Management. https://doi.org/10.1108/JFBM-02-2021-0013

Hosseini,E; Tajpour, M; Salamzadeh,A; Demiryurek,K & Kawamorita,H(2021), Resilience and Knowledge-Based Firms’ Performance: The Mediating Role of Entrepreneurial Thinking. Journal of Entrepreneurship and Business Resilience, 4(2), 7-29.

Response 6: Thank you very much for your references, which have provided important help for the improvement of this paper. We have carefully considered their convergence with this paper and quoted specific suggestions on the construction of open source community in the conclusion.

First paper let us in the open source community material subsidies, supplement the importance of spiritual enterprises trust, more fully expounds the direction of the open source community building.

The second paper for the open source community has provided valuable advice on the specific management, such as creating a database of open source participants results and information system, promoting the value of innovation Consolidate the sustainable relationship between the enterprise and its partners.

  1. Tajpour, M., Salamzadeh, A., Salamzadeh, Y., & Braga, V. (2021). Investigating social capital, trust and commitment in family business: case of media firms. Journal of Family Business Management. https://doi.org/10.1108/JFBM-02-2021-0013
  2. Hosseini,E; Tajpour,M; Salamzadeh,A; Demiryurek,K & Kawamorita,H(2021), Resilience and Knowledge-Based Firms’Performance: The Mediating Role of Entrepreneurial Thinking. Journal of Entrepreneurship and Business Resilience, 4(2), 7-29.

Reviewer 3 Report

I found the paper very well and very suitable for the journal because of its relevance to the aim and scope of the journal.  

Author Response

Point 1: I found the paper very well and very suitable for the journal because of its relevance to the aim and scope of the journal.

Response 1: Thank you very much for your approval.

Round 2

Reviewer 1 Report

Thank you for revising this interesting article. In my opinion, you have taken into account all the short comings I mentioned. This with the other additions made have improved the quality of the proposed article significantly. Therefore, I think this article is suitable to be considered for publication.

Reviewer 2 Report

Dear author(s)

Hope you are doing well. According to the review of this article, the corrections have been made.

Good luck